# Purine Auxotrophic Starvation Evokes Phenotype Similar to Stationary Phase Cells in Budding Yeast

**DOI:** 10.3390/jof8010029

**Published:** 2021-12-29

**Authors:** Agnese Kokina, Kristel Tanilas, Zane Ozolina, Karlis Pleiko, Karlis Shvirksts, Ilze Vamza, Janis Liepins

**Affiliations:** 1Institute of Microbiology and Biotechnology, University of Latvia, Jelgavas 1, LV-1004 Riga, Latvia; zane.ozolina2@lu.lv (Z.O.); karlis.svirksts@lu.lv (K.S.); ilze.vamza@gmail.com (I.V.); janis.liepins@lu.lv (J.L.); 2Center of Food and Fermentation Technologies, Akadeemia Tee 15A, 12618 Tallinn, Estonia; kristel@tftak.eu; 3Faculty of Medicine, University of Latvia, Jelgavas 3, LV-1004 Riga, Latvia; karlis.pleiko@lu.lv; 4Laboratory of Precision and Nanomedicine, Institute of Biomedicine and Translational Medicine, University of Tartu, 50411 Tartu, Estonia

**Keywords:** *Saccharomyces cerevisiae*, starvation, purines, stress resistance

## Abstract

Purine auxotrophy is an abundant trait among eukaryotic parasites and a typical marker for many budding yeast strains. Supplementation with an additional purine source (such as adenine) is necessary to cultivate these strains. If not supplied in adequate amounts, purine starvation sets in. We explored purine starvation effects in a model organism, a budding yeast *Saccharomyces cerevisiae* *ade8* knockout, at the level of cellular morphology, central carbon metabolism, and global transcriptome. We observed that purine-starved cells stopped their cycle in G1/G0 state and accumulated trehalose, and the intracellular concentration of AXP decreased, but adenylate charge remained stable. Cells became tolerant to severe environmental stresses. Intracellular RNA concentration decreased, and massive downregulation of ribosomal biosynthesis genes occurred. We proved that the expression of new proteins during purine starvation is critical for cells to attain stress tolerance phenotype Msn2/4p targets are upregulated in purine-starved cells when compared to cells cultivated in purine-rich media. The overall transcriptomic response to purine starvation resembles that of stationary phase cells. Our results demonstrate that the induction of a strong stress resistance phenotype in budding yeast can be caused not only by natural starvation, but also starvation for metabolic intermediates, such as purines.

## 1. Introduction

Budding yeast *Saccharomyces cerevisiae* is a unicellular fungus that has evolved for fast growth in the presence of abundant nutrients [1]. Deprivation of certain nutrients could be interpreted as a prelude to potentially more serious stressors to come. A lack of nutrients drives cells to enter a stationary phase, in which they do not proliferate, but become more stress resistant. In the case of carbon, nitrogen, or phosphorus depletion (so-called natural starvation, as yeast cells can experience these in the wild), yeast cells stop the cell cycle and activate specific gene expression patterns, thus forming a general stress-resistance phenotype [2,3,4]. In large genome-scale phenotyping screens, many mutations have been identified that increase budding yeast fitness when they are starved of carbon or nitrogen [5].

When setting up cultivation experiments, it is often important to reach an appropriate growth rate and biomass yield and avoid unexpected phenotypic side effects. Therefore, proper concentrations of all nutrients in the media should be ensured [6]. Alternatively, the sudden depletion of nutrients could initiate nutrient-specific starvation and thus induce a number of phenotypic effects in yeast cells [7,8,9,10]. Due to introduced auxotrophic markers in laboratory strains, specific “synthetic starvation” could set in when a specific auxotrophic nutrient is not supplied or is exhausted. Gomes and colleagues show that limiting essential auxotrophic amino acids decreases the final biomass yield and stress resistance of yeast [11]. In the case of uracil or leucine starvation, cells fail to enter the stationary phase and mostly die in the exponential growth phase [3,9]. On the other hand, methionine starvation has shown signs similar to the natural starvation response [8,12], which is explained by methionine being a source of sulfur. This shows that not all auxotrophies are the same, and starvation for these nutrients can have dramatically different effects on yeast metabolism. In addition, the genetic background of the strain can affect the phenotypic response to environmental changes [5,11,13,14,15]. Therefore, to continue exploiting model yeast strains in fundamental or applied research, detailed knowledge on their physiology in every possible environmental or laboratory setting is invaluable.

Although adenine auxotrophic strains are widely used, their phenotypic response to purine starvation has not been studied in detail. Purine is a ubiquitous molecule in the cell and forms DNA, RNA (adenine and guanine nucleotides), and cofactors (NAD, FAD). Purine synthesis is highly conserved among eukaryotes. In budding yeast, it consists of a linear chain of 10 sequential reactions coded by *ADE1*/*2*/*4*/*5*,*7*/*6*/*8*/*12*/*16*,*17* genes. This pathway produces inosine monophosphate (IMP), which is a branching point to adenine and guanine. *ADE8* codes for phosphoribosylformylglycinamidine synthase, which is the third enzyme in the chain. Currently, no specific regulation activity of *ADE8* on IMP production is known [16]. If *ade8* mutant is placed in adenine-deficient media, it cannot produce IMP, thus neither adenine nor guanine is produced, and purine auxotrophic starvation sets in.

Until now, we have demonstrated some purine starvation effects on *ade2* strain in the W303 strain background. In the case of purine starvation, W303 *ade2* became desiccation tolerant, the budding index decreased, and trehalose content increased [17].

In this study we examined the global effects caused by purine starvation in *ade8* knockout in CEN.PK2-1D strain. We analyzed various aspects of the cell phenotype: cell growth, cell cycle state, changes in central carbon metabolism, cell ATP content, sublethal stress resistance, and genome-wide transcriptomic response. Our results imply that purine auxotrophic starvation initiates the formation of a stress-resistance phenotype and reroutes the energetic metabolism toward fermentative growth. The transcription pattern of purine-starved cells resembles that of stationary phase cells starving for carbon.

## 2. Materials and Methods

### 2.1. Strains and Cultivation Conditions

Wild-type strain CEN.PK2-1D *MATalpha his3Δ1*; *leu2-3_112*; *ura3-52*; *trp1-289*; *MAL2-8c*; *SUC2* was a gift from Peter Richard, VTT Biotechnology, Finland. The *ade8* CEN.PK2-1D *ade8Δ0*. *ade8* knockout was induced by the *ura3-URA3* 5-FOA toxicity knockout technique, using *ade8* knockout construct plasmid from [18]. Cultures were maintained on YPD agar and kept at 4 °C. Fresh YPD agar plates were regularly reinoculated from stock cultures kept at −80 °C.

Strains were cultivated in Synthetic Defined (SD) media [19] with 80 mg tryptophan, 100 mg uracil, 480 mg leucine, 100 mg histidine, and 100 mg adenine added per liter, as suggested in [6]. To ensure that yeast cultures were in the exponential growth phase, we reinoculated overnight cultures (grown from a single colony) into fresh media, where at least 6 doublings occurred and OD_600_ 0.5–1, corresponding to 1–2∙10^7^ cells mL^−1^, was reached. Cultures in the exponential growth phase (OD 0.5–1) were washed with distilled water twice and resuspended at OD 0.5 in full SD media (SD) or SD media with adenine omitted (SD ade−).

All cultures were incubated on a rotary shaker at 30 °C and 180 rpm. To demonstrate changes in optical density during starvation, 96-well Tecan Infinite M200 multimode reader was used with the following cultivation cycle: orbital (3.5 mm) shaking for 490 s, waiting for 60 s, optical density measurement at 600 nm. Alternatively, culture growth dynamics was measured with a Z2 Cell and Particle Counter (Beckman Coulter, Brea, CA, USA).

### 2.2. NMR Analysis of Extracellular Amino Acids and Purines

Cell-free culture media was mixed with DSS (sodium 4,4-dimethyl-4-silapentane sulfonate) in D_2_O to obtain a final DSS internal standard concentration of 1.1 mM and transferred to a 5 mm NMR sample tube. NMR analysis was performed at 25 °C on a 600 MHz Bruker Avance Neo spectrometer equipped with a QCI quadruple resonance cryoprobe. The noesypr1d pulse sequence was used with water suppression during a recycle delay of 10 s. The spectral width was 11.9 ppm, and 128 scans were collected into 32K data points using an acquisition time of 2.3 s. The acquired 1H NMR spectra were zero-filled once, and no apodization functions were applied prior to Fourier transformation. Phase and baseline corrections were applied manually. Spectra were referenced to DSS (at 0.00 ppm). The identification and quantification of sample components were performed using Chenomx NMR Suite professional software (version 5.11; Chenomx Inc., Edmonton, AB, Canada).

### 2.3. Cell Morphology Measurements

Cell samples before and after 4-h cultivation in media with (SD) or without adenine (SD ade−) were fixed in formaldehyde 0.5% and examined with an optical microscope (Olympus BX51, Tokyo, Japan). Microphotographs (1360 × 1024 pixels) were obtained with a digital camera (Olympus DP71, Tokyo, Japan). Cell size and budding index were determined by microphotography analysis in the ImageJ program. Budding index was defined as the proportion between the number of cells with buds and the total cell number. Bud was defined as a cell with a cross-section area less than half the mother cell size. Cell size was determined as the cell cross-section area measured from the microphotographs using ImageJ. Cells were defined as ellipses, with area measured in pixels and recalculated to square micrometers (1 μm = 5.7 pixels). For each sample, at least 500 cells were measured.

### 2.4. Flow Cytometry

Cell DNA content was determined by flow cytometry as described in [20]. Briefly, 0.5 mL of yeast culture was fixed in 10 mL of ice-cold 70% ethanol for at least 15 min and washed once with 50 mM citric acid. RNA was degraded using RNase A (10 μg mL^−1^) in 50 mM citric acid overnight at 37 °C. DNA was stained with 10× SYBR Green (Invitrogen, Waltham, MA, USA) in 50 mM citric acid for 30 min. Cells were analyzed with a FACSAria device (Becton Dickinson, Franklin Lakes, NJ, USA). Cell cycle distribution was analyzed with Cyflogic software.

### 2.5. Fermentation and Metabolite Flux Measurements

Fermentation was done in a Sartorius Q-plus fermentation system with working volume of 0.3 L, gas flow 0.25 L·min^−1^, mixing rate 400 rpm, media pH set to pH 5.5. Biomass concentration was determined as absorbance in 590 nm (WPA Colorimeter Colourwave CO7500, Biochrom, Cambridge, UK). The following coefficient to convert absorbance units to dry weight was used: 1 OD_590_ = 0.278 g∙L^−1^. Carbon dioxide evolution was recorded by an exhaust gas analyzer (Infors Gas Analyser, InforsHT, Basel, Switzerland) in parallel with harvesting metabolite samples.

The contents of extracellular glucose, ethanol, acetate, and glycerol were measured simultaneously by an Agilent 1100 HPLC system with a Shodex Asahipak SH1011 column, and they were quantified with a refractive index detector (RI detector RID G1362A). The flow rate of the mobile phase (0.01 N H_2_SO_4_) was 0.6 mL min^−1^ and the sample injection volume was 5 μL. Biomass from fermentations was centrifuged and intracellular nucleotide pools were extracted via cold methanol extraction. ATP, ADP, and AMP were quantified by HPLC-MS-TOF analysis, as described in [21].

### 2.6. FTIR Analysis

For cell macromolecular content analysis, Fourier-transform infrared (FTIR) spectroscopy was used as described in [22]. For this analysis, 2 mL of cells (OD_600_1–4) was harvested by centrifugation and washed 3 times with distilled water. Cell pellets were diluted with 50 μL of distilled water, and samples were spotted on 96-well spot-plates. Absorbance data were recorded by a Vertex 70 device with HTS-XT microplate extender, interval 4000–600 cm^−1^, resolution 4 cm^−1^. For data collection and control, OPUS/LAB 6.5 software was used.

### 2.7. Cell Carbohydrate Extraction and Quantification

Fractional cell polysaccharide purification for quantitative assays was done as described in [23]. Total carbohydrate content of each fraction was determined by anthrone assay, and results were expressed as glucose equivalent mg∙gDW^−1^ biomass [24].

### 2.8. Transcriptomics

Total yeast RNA after 4-h cultivation in synthetic dextrose (SD) or SD media with adenine omitted (SD ade−) was isolated with a RiboPure™ RNA Purification Kit for yeast (Thermo Scientific, Waltham, MA, USA). RNA samples for each condition were harvested in triplicate. Cell pellets from 50 mL suspensions were frozen in liquid N_2_ and stored at −80 °C. RNA samples were prepared using 3′ mRNA-Seq Library Prep Kit (Lexogen, Vienna, Austria) according to the manufacturer’s protocol. Yeast transcriptome was analyzed using MiSeq (Illumina, San Diego, CA, USA) NGS data analysis. Sequencing reads were quality filtered (Q = 30), Illumina adapters and poly-A tails were removed, and reads at least 100 nt in length were selected for further processing using cutadapt (see Appendix A for details). S288c reference genome from yeastgenome.org was used to identify gene transcripts.

Genes with lower than 1 count per million (CPM) in fewer than 2 samples were filtered out. The Benjamini and Hochberg method was used to calculate multiple comparison adjusted *p*-value as false discovery rate (FDR). FDR < 0.001 with logFC > 2 was set as a threshold for significance. Expression data set were submitted to the European Nucleotide Archive (ENA) database, under accession no. PRJEB40525.

### 2.9. Sublethal Stresses

Cells were grown in SD media until the exponential phase, washed with distilled water twice, and inoculated in SD or SD ade− with cell density of 1 × 10^7^ cells∙mL^−1^. After 4-h incubation, cells were harvested by centrifugation, washed with distilled water once, and aliquoted in 1 mL, with OD_600_ = 1 (corresponding to 2 × 10^7^ cells∙mL^−1^). Three aliquots were exposed to each stress. For thermal stress, cells were kept at 53 °C for 10 min. For oxidative stress, cells were incubated in 10 mM H_2_O_2_ for 50 min, then washed with distilled water. For desiccation, cells were sedimented by centrifugation, the supernatant was removed, and the pellet was air-dried in the desiccator at 30 °C for 6 h. After drying, distilled water was added to resuspend cells. After all stress treatments, cells were serially diluted, and dilutions were spotted on YPD plates to assess CFU∙mL^−1^. To check for cell loss during washing steps, the OD of the suspension was measured and CFU∙mL^−1^ corrected for OD value. Survival is expressed as % assuming that OD_600_ = 1 corresponds to 2∙10^7^ cells∙mL^−1^. To test weak acid stress resistance, cells were spotted on YPD plates supplemented with 0.1 M acetic acid, with pH of agar media set to 4.5 [25].

## 3. Results

To characterize global changes initiated by purine starvation, we constructed *ade8* knockout in laboratory yeast strain CEN.PK2-1D background (*ade8* strain). To investigate the physiological effects imposed by purine starvation in this background, we cultivated *ade8* strain in SD media with all necessary auxotrophic supplements present in surplus media (SD) and media with adenine omitted (SD ade−).

### 3.1. Growth of CEN.PK2-1D ade8 Ceases in the Absence of Purine

CEN.PK2-1D strain, which is often used in laboratory experiments, contains several auxotrophic markers: histidine (*his3*), leucine (*leu2*), tryptophan (*trp1*), and uracil (*ura3-52*). We introduced additional purine auxotrophy by “clean” (antibiotic marker-free) *ade8* knockout, as suggested by [18] The lack of any single supplement necessary to complement the metabolic needs of this strain leads to growth cessation (Figure 1a). When exponentially growing CEN.PK2-1D *ade8* cells were washed with distilled water and inoculated in SD ade− media, increased cell numbers were observed in the first two hours, but then cell numbers (mL^−1^) remained stable (Figure 1b). For most of our experiments, we chose cells that had been starved in SD ade− for 4 h, as that is the time when ade− specific phenotype appears. To ensure that missing purine is the only factor influencing cell phenotype, we made sure that all other auxotrophic agents were still in media after 4 h of cultivation in SD or purine starvation media (Figure 1c).

Therefore, we conclude that our media composition can induce starvation specifically for adenine (purine), and the subsequent physiological effects observed are solely due to the lack of an external purine supply.

### 3.2. Cell Cycle Arrests in G1/0 during Purine Starvation

Although cell number∙mL^−1^ did not increase after the second hour of *ade8* cultivation in purine starvation media, elevated optical density over time was observed (Figure 1a,b). This led us to hypothesize that specific changes in cell morphology occur, increasing light dissipation and accounting for increase OD during *ade8* purine starvation. We quantified the budding index, analyzed the DNA content of the cell using FACS, and measured cross-sections of cells of the *ade8* grown in SD and SD ade− media.

The budding index is an indicator of culture progression through the cell cycle. We defined the budding index as the ratio (%) of the number of cells with buds and the total number of cells (see Figure 2a).

We observed that the budding index was approximately 30% when cells were cultivated in SD media, while in purine starvation media it significantly decreased (down to 15%). To complement budding index data and test DNA content in the cells of a population grown in SD or SD ade− media, we performed FACS analysis. We found that the purine starvation culture became enriched with cells harboring N copies of DNA per cell, which occurred within the first 2 h of cultivation. After 4 h, the *ade8* cell population, when cultivated in SD media, contained both N and 2N DNA copies per cell, while cells cultivated in SD ade− contained mostly N copies of DNA (comparison shown in Figure 2b).

We tested whether cell size changes could contribute to OD increase during purine starvation. We measured the average cell cross-section after 4 h of cultivation in SD and SD ade− media. Indeed, purine-starved *ade8* cells were larger than cells growing in SD media, as shown in Figure 2c. Therefore, we conclude that purine starvation leads to a drop in the budding index accompanied by increased cell population of cells with N copies of DNA and significantly increased size (as demonstrated by increased cross-section). These morphological markers demonstrate that purine-starved cells are morphologically different from cells growing in SD media.

### 3.3. Purine Starvation Slows Glycolysis

Changes in culture growth parameters (optical density or cell concentration) are probably the most obvious phenotypic markers of auxotrophic starvation. To further investigate purine starvation effects, we measured several metabolic markers when *ade8* was cultivated in SD or SD ade− media. We measured glucose consumption and production of major carbon metabolites (ethanol, CO_2_, glycerol, acetate, and biomass) and determined intracellular ATP, ADP, and AMP concentrations.

The specific growth rate of *ade8* in SD media was 0.4 h^−1^ and in SD ade− media was 0.15 h^−1^. The glucose-specific uptake rate q during purine starvation was two times smaller than in SD media: 43 +/−3.6 and 81+/−5.4 mCMol ∙gDW^−1^∙h^−1^, respectively. Meanwhile, the specific CO_2_ production rate was five to six times higher in SD media than in SD ade− media (see Figure 3b). Since glucose was the sole carbon source and we could account for more than 90% carbon in total, we calculated flux distribution as carbon % of glucose consumed for *ade8* cultivated in SD or SD ade− (see Figure 3a and raw flux data in Appendix A). Part of the carbon was rerouted away from biomass growth, and glycerol and acetate accumulated instead. Additionally, amount of CO_2_ released from the purine-starved cells was equimolar to the ethanol produced. This, in turn, means that other pathways where CO_2_ is produced (pentose phosphate pathway, mitochondria TCA) might be suppressed in purine-starved cells.

When cell growth is suspended due to the lack of an essential metabolite (purine), not only are main carbon fluxes affected (Figure 3a,b), but so are concentrations of intracellular purine nucleotides. We measured the concentrations of purine-containing moieties (ATP, ADP, and AMP) to find out if the intracellular concentration of these molecules changed if the external supply of precursor adenine was diminished (see Figure 3c).

Indeed, already 1.5 h after shifting the SD media to SD ade−, the intracellular concentrations of ADP and ATP dropped more than half of the initial values (Figure 3c). Interestingly, while intracellular concentrations of ATP, ADP, and AMP dropped significantly during purine starvation, energy charge throughout purine starvation remained almost constant. There was a slight drop in the beginning of starvation, but adenylate charge reached pre-starvation levels in the cells after that. It should be kept in mind that in the first two hours of purine starvation, cells were still proliferating (see Figure 1b).

Although *ade8* growth in SD ade− media stopped, cells continued to metabolize glucose. However, specific glucose uptake dropped significantly, from 81+/−5 to 43 +/−3 mCmol ∙g DW∙h^−1^. We think this is related to the decreased intracellular adenine nucleotide concentration (Figure 3c), which does not allow rapid glucose metabolism [26]. The distribution of other carbon fluxes was also altered. In SD media, most of the energetic needs seem to be fulfilled with the help of fermentation; still, there is also CO_2_ production that does not come from ethanol production, which points to the involvement of mitochondrial activity and respiro-fermentative growth. In SD ade− conditions, all CO_2_ can be attributed to ethanol production. Interestingly, glycerol production significantly increased.

It seems that instead of biomass, a significant amount of carbon is redirected to glycerol synthesis, which points to an increase of cell glycerol and/or lipid content, as glycerol is the backbone of triacylglycerols (TAGs).

We checked whether the macromolecule content of the biomass was affected when cells were cultivated in SD media or purine starved. Relative amounts of proteins and nucleic acids decreased, while carbohydrates and lipids increased (see Figure 4a).

We extracted fractions of the main budding yeast reserve carbohydrates, trehalose, and glycogen, and quantified the carbohydrate content of each fraction by the anthrone method. The results show that during 4 h of purine starvation, *ade8* cells accumulated 100 mg trehalose and 263 mg glycogen per g DW, while cells growing in SD media had 17 mg and 102 mg, respectively. We also measured the concentrations of other cellular carbohydrates that make up most of the yeast biomass, mannans, and beta-glucans. Increased carbohydrate fraction during purine starvation is due to the accumulation of reserve carbohydrates, while the amount of structural carbohydrates does not change (Figure 4b). Purine-starved yeast biomass accumulated more reserve carbohydrates by 244 mg∙gDW^−1^, and total carbohydrate content increased by 227 mg∙gDW^−1^, which correlates to an increased carbohydrate fraction in the biomass macromolecular composition. The carbohydrate fraction in purine-starved cell biomass almost doubled, as revealed by FTIR analysis (see Figure 4a).

To verify the biomass macromolecular composition data (Figure 4a), we estimated the RNA content of the cells. We assessed RNA quality (Qiaxcell capillary electrophoresis) before expression analysis by comparing the 18S and 28S rRNA ratio. The ratio between 18S and 28S did not change (whether it was an SD or ade− sample), therefore RNA quality was good, and no degradation was observed. At the same time, the amount of extracted RNA per unit of biomass was three times smaller in adenine-starved cells, thus reinforcing the biomass macromolecular content obtained by FTIR: decreased nucleic acid amount during purine starvation.

### 3.4. Purine Starvation Elicits Strong Stress Resilience

Carbon flux distribution away from energy production toward storage metabolites increased glycerol production, decreased glucose uptake, and decreased intracellular adenine nucleotide content. This indicates that energy and carbon in purine starvation might be redirected to other functions, away from biomass synthesis and growth.

Stress resistance is an important phenotypic feature of microbial cells. It is known that the general environmental stress resistance (ESR) phenotype is induced in slowly growing, stationary, or quiescent cells (reviewed in [27]). Previously, it was shown that methionine auxotrophic starvation can lead to elevated stress resistance [8]. To test if cultivation in purine starvation media would affect culture stress resistance, we tested cell viability after exposure to harsh environmental stresses (sublethal stress resistance). We tested whether purine-starved cells would have higher resistance to short thermic or oxidative stress and weak acid stress (growth on plates containing acetic acid at pH 4.5, to see long-term stress resistance). Additionally, we tested desiccation tolerance as an example of a multicomponent stressor. The time of exposure of each stress was adjusted so that the survival of exponentially growing cells would be approximately 10%.

We observed that the survival of purine-starved cells was more than 10-fold higher than that of the cell population growing in SD media in all sublethal stresses tested (see Figure 5a).

We wanted to understand whether desiccation tolerance is a phenotype that evolves immediately after the shift to starvation media, or if some “adaptive reactions” occur while cells are starving for purine. To test this development of the stress resistance phenotype, we used a desiccation assay, since this treatment gave the most distinct signal for cells cultivated in SD ade− and SD media. To determine that, we added cycloheximide translation inhibitor to cells either at the beginning of starvation or 2 h after, when active cell proliferation had ceased. The results are depicted in Figure 5b. Translation inhibition during auxotrophic starvation lowered desiccation tolerance after starvation. Therefore, we conclude that the stress resistant phenotype indeed develops during purine starvation via new protein production, and signaling of the lack of purine also starts while cells are performing their final division.

Our results demonstrate that purine starvation preconditions cells to become stress resistant. However, it is not known how cells coordinate the lack of purine with these massive phenotypical changes. Other authors have shown that desiccation tolerance can be TOR pathway dependent [28]. This fact, together with the decreased RNA content, led us to inquire whether the TOR signaling system mediates purine starvation toward the development of a specific phenotype. We used desiccation tolerance as a marker of the purine starvation phenotype and compared the desiccation tolerance of purine-starved cells and cells grown in SD media, adding rapamycin during incubation, in an experimental setup similar to the cycloheximide assay. Rapamycin inhibits downstream signaling from target of rapamycin (TOR) proteins. The addition of rapamycin did affect desiccation tolerance, but to a lesser extent than purine starvation (see Figure 5b). For purine-starved cells, 19% of cells were desiccation tolerant if no rapamycin was added, compared to 12% after 2 h of rapamycin treatment and 1% after 4 h of rapamycin treatment. The addition of rapamycin to SD grown cells increased their desiccation tolerance, but not to the degree of purine-starved cells: 0.05% with no rapamycin, 1% with 2 h rapamycin, and 1% with 4 h rapamycin (see Figure 5b). Since purine-starved cells exhibit 10–20 times higher desiccation tolerance than rapamycin-treated cells, we conclude that the TOR system might be involved in purine starvation signaling, but there are additional systems in play.

### 3.5. Purine-Starved Cells Exhibit a Distinct Transcriptome Resembling Stationary Phase Cells

The results from translation inhibition (cycloheximide assay, Figure 5b) show that stress resistance is driven by gene expression. Therefore, to assess gene expression changes over purine starvation, we performed transcriptome analysis via RNAseq.

RNA was extracted from flash-frozen yeast biomass from *ade8* cells that spent 4 h in either SD or adenine-deficient media. The samples were harvested in biological triplicate and all data represent the average of the triplicates. The significance criterion for gene up- or downregulation was chosen as the logarithm of fold change less than −2 (downregulated) or greater than 2 (upregulated). When comparing expression data of *ade8* cells cultivated in SD ade− or SD, we found 455 significantly upregulated genes (more expressed in SD ade− conditions) and 244 downregulated genes (more expressed in SD). The transcription of the rest of the genes was not significantly affected by the presence or absence of purine. Among the top 20 most upregulated transcripts, we found genes coding for stress resistance proteins (*SIP18* and its paralogue *GRE1*, *DDR2*, *HSP12*, *HSP26*), stationary phase response proteins (*SPG4*, *SPG1*) and carbohydrate metabolism (*HXT5*, *HXT6*, *TKL2*, *GND2*). Interestingly, expression of several spore-related genes was also upregulated, for example *SPS100*. For a full gene list, see Appendix A.

The most prominent downregulation was observed in the expression of various tRNA and protein genes related to the transcription process. This was also demonstrated by a GO term enrichment search for all genes that were affected by purine starvation (−2 > logFC > 2). The results are depicted in Figure 6. Metabolic process enrichment terms were grouped in two clusters corresponding to up- and downregulated genes: one cluster including genes of redox processes, and various catabolic genes, which were upregulated, and the other cluster consisting of genes mainly connected with translation, which were downregulated (Figure 6a). The metabolic processes that were the most significantly enriched in our dataset were connected with downregulated genes and the translation process. From the upregulated genes, redox processes and carbohydrate metabolism were the GO terms that were most enriched in purine-starved cells. A table with all GO terms and their expression statistics can be found in Appendix A.

When analyzing GO terms for enriched genes, we saw several terms connected to metabolism, so we looked into the metabolic genes with regard to which pathways were affected and whether the results coincided with our metabolite analysis. By plotting all gene expression changes on the yeast metabolic pathway (also below log_2_FC 2/−2) (https://pathway.yeastgenome.org/), we could observe that genes of energetic metabolism were mainly upregulated in glycolysis (*GXK1*, *HXK1*, *TDH1*, *ENO1*, *PYK2*); also, Krebs cycle and glyoxylate cycle genes and their isoforms were more highly expressed compared to fast-growing cells, with the exception of *ACO2* and *MAE1*, which were less expressed. At the same time, it can be seen that genes involved in the use of alternative carbon sources were also upregulated: galactose (*GAL10*, *GAL1*) and xylose (*GRE3*, *XYL2*). In the electron transport chain, we see that genes for NADH dehydrogenase (*NDI1*), all genes for both isoforms of succinate dehydrogenase (*SDH4*, *SDH3*, *SDH2*, *SDH1*, *YJL045W*), and some genes of ubiquinol cytochrome c reductase complex (*QCR9*, *QCR10*) were also upregulated.

It appears that although the ATP charge is not changed, a considerable drop in ATP concentration induces energy production pathways to gain more ATP. This also applies to less commonly used AcetylCoA sources; we can see genes of fatty acid oxidation being upregulated (*POX1*, *FOX2*, *POT1*), along with acetate utilization (*ACS1*) and ethanol degradation (*ADH2*, *ALD2*, *ACS1*). Although all of these genes are expressed in higher amounts, which points to increased flux through energetic metabolism and Krebs cycle, cells do not produce more CO_2_ or have increased glucose consumption. It looks like purine starvation causes dysregulation of carbon energetic metabolism, where cells are trying to achieve a higher concentration of ATP, by using all available carbon sources and removing glucose repression during metabolism.

At the same time, we see an accumulation of reserve carbohydrates that is not reflected in gene expression. We do observe an accumulation of trehalose and glycogen, but gene expression levels in the pathways for degradation and synthesis are upregulated for trehalose and glycogen in a similar manner. Some gluconeogenesis genes are also upregulated (*MDH2*, *PCK1*).

Several amino acid related processes are affected. Besides purine auxotrophy, *ade8* strain is an auxotroph for tryptophan, histidine, leucine, and uracil. Indeed, amino acid Trp, Leu, and His synthesis pathways are downregulated due to the adequate supply of respective amino acids from the medium. None of these amino acids get depleted during purine starvation (see Figure 1c). However, we see also downregulation of other amino acid synthesis: Tyr, Phe, Met, Ser, Pro, Asn, Arg, Val, Ile, and Lys pathways, which agrees with our observations on growth cessation and the downregulation of translation processes. At the same time, production of glutamate from -oxoglutarate and glycine from glyoxylate are upregulated, which would explain the observed upregulation of Krebs cycle and glyoxylate cycle enzymes.

Several cofactor metabolism pathways were affected with oxidative part of PPP and 1C metabolism most prominently. In PPP, *SOL4* and *GND2* genes are highly upregulated, and in 1C metabolism most of the genes are downregulated with the exception of glycine cleavage complex, which provides 5,10 methylenetetrahydrofolate.

Our strain was defective in *ade8* gene, which is the third step of purine biosynthesis that starts with PRPP, and these three steps require glutamine, glycine, ATP, and 10-formyltetrahydrofolate as cofactors. We can say that we noted genetic upregulation of processes providing these substances. However, as purines and the cofactors that are produced for synthesis are involved in a variety of cellular processes, metabolic dysregulation happens on several levels.

We compared the transcriptome of *ade8* during purine starvation with that of stationary phase cells (Geodataset GSE111056 [30]). We chose to compare our dataset with the data acquired using the Illumina sequencing platform (similar to ours) with at least n = 2 replicates for each condition from GEO. We used the same data handling procedure as with our expression data (as described in Appendix A). When comparing the two datasets, we found that similar genes were simultaneously upregulated and downregulated in both conditions (see Figure 7 and a full list of gene expression analyses in Appendix A). When plotted on a cell map, downregulation of genes associated with translation is most obvious (Figure 7b). On the other hand, upregulated genes do not show such clear clustering. Genes related to peroxisome function seem to be upregulated in both datasets. We performed an analysis of GO term enrichment within genes that were upregulated in purine-starved and stationary phase cells to see if there were some common functions (Figure 7c). GO terms that were enriched in genes that were upregulated in both conditions were similar to the GO terms of purine-starved cells; mainly genes connected to catabolism, reserve carbohydrates, and redox processes were found to be enriched in this dataset.

## 4. Discussion

Purine auxotrophic starvation induces a phenotype distinctive from exponential cells. Yeast cells halt proliferation, stop cell cycle in G1/0, and accumulate reserve carbohydrates (glycogen and trehalose) to become resilient to multiple stresses. In addition, purine starvation elicited a transcription pattern distinct from exponential cells, sharing many traits with stationary phase cells. Currently, it is not known whether these remarkable phenotypic changes are induced by purine depletion per se, or lack of purine metabolites initiates effects similar to “natural” starvation (nitrogen or carbon), because purine depletion might be signaled through the same pathways as natural starvation. We will discuss potential scenarios showing how purine depletion can lead to the specific purine starvation phenotype we observed.

### 4.1. Intracellular Adenylate Pool Is Not Sufficient to Sustain Cell Proliferation

Purines are essential metabolites of the cell metabolism. Growing cells metabolize purines (guanine and adenine) to fuel the synthesis of new DNA and RNA nucleotides. Additionally, purines are used as cofactors and energy-carrying substances in myriad catabolic and anabolic reactions. For example, ATP and GTP are necessary for translation to occur; two ATP molecules are invested to activate each glucose molecule within glycolysis, etc.

The specific ATP consumption of 1 g of growing CEN.PK strain cell biomass is approximately 5.7 mM g^−1^DW h^−1^. A significant portion of that (70%, or at least 4.5 mM ATP g^−1^DW^−1^ h^−1^) is devoted to protein synthesis and turnover [31]. Approximately 0.63 mM ATP g^−1^DW^−1^ h is used for maintenance functions during aerobic cultivation [32]. Thus, it is possible to keep cells alive in the nonproliferative state with a handful of ATP, as we have seen in purine-starved cells.

Purine concentration within the cytoplasm of budding yeast is about 7 mM, most of which is ATP (5 mM) and GTP (1.5 mM) [33]. Meanwhile, the purine content of the haploid genome of budding yeast is 4.65 × 10^6^ of guanine and 7.45 × 10^6^ adenine nucleotides, which is approximately 1/10 of the molar amount of purines in the cytoplasm. Exponentially growing yeast cells contain approximately 50 times more RNA than DNA [34]. Therefore, when the external supply of purine is stopped, cytoplasmic purine resources alone cannot ensure the needs of new daughter cells in purine auxotrophic cells. However, purine auxotrophs tend to accumulate purine moieties in the form of inosine or hypoxanthine [17,35] within vacuoles of their cells. The existence of purine reserves accumulated within the cells of purine auxotrophs can explain the increased cell number during the first hours of cultivation in purine-deficient medium (Figure 1b). We observed that cells had a lower amount of rRNA after 4 h of starvation. After the beginning of purine starvation, most yeast cells finished the DNA synthesis phase and halted budding; putative arrest in G1 was observed (see Figure 1b and Figure 2b). Most of a cell’s RNA is ribosomal RNA (rRNA), therefore, by degrading rRNA it would be possible to free nucleotides required for DNA synthesis. Analyzing the expression data, we see that the expression of ribonucleotide reductase *RNR2* (log_2_FC = 2.04) and RNR4 (log_2_FC = 1.64) was upregulated after 4 h in ade− media. The observed replication cessation indicates that auxotrophic cell growth strongly depends on an external purine supply. If purine supply stops, then the internal purine reserves of auxotrophic cells cannot sustain further proliferation and cell doubling ceases; internal reserves may sustain one doubling, but no more.

### 4.2. Purine Starvation Induces Accumulation of Metabolites Capable of Increasing Stress Tolerance

If mild stress is applied, cells adapt to it and become ready for stronger challenges in the future. For example, NaCl pre-treatment increases yeast cell tolerance to H_2_O_2_ stress [36]. Additionally, it was demonstrated that methionine starvation might also prepare cells to be resilient to peroxide stress [8]. Previously, we showed that desiccation tolerance of W303*ade2* cells was indeed significantly higher if cells were starved for purine [17]. Here we explored whether purine starvation “prepares” cells for multiple sublethal stresses (not only desiccation), and, indeed, we found that a strong, resilient phenotype is generated during purine starvation. Moreover, if protein expression is blocked by cycloheximide during purine starvation, the formation of a resilient phenotype is abolished (Figure 5b, cycloheximide treatment). This, in turn, points out that the resilience phenotype is established due to the expression of specific genes or induction of a genome-wide transcriptional program.

When cultivating *ade8* strain in completely synthetic media, the main fermentation products were ethanol, biomass, and CO_2_, but in the purine-deficient media acetate and glycerol accumulated, while the carbon proportion devoted to biomass decreased (see Figure 3b). Reserve carbohydrates (trehalose and glycogen) also accumulated in the purine-starved cells (Figure 4b). Interestingly, the accumulation of glycerol and/or trehalose per se is attributed to increased stress tolerance and can help in survival after strong environmental perturbations (desiccation, extreme heat, etc.) [37,38]. Besides small carbon metabolites (e.g., trehalose), other factors can ensure viability after desiccation. Heat shock proteins (Hsp12p, Hsp26) are the next most important factors after trehalose that can ensure tolerance to multiple stresses, including desiccation [39]. Indeed, the expression of *HSP12* during purine starvation was highly upregulated (see Appendix A). In fact, it was among the top 10 most upregulated transcripts. Therefore, accumulation of trehalose and Hsp12p can explain the high stress tolerance of purine-starved cells.

### 4.3. Msn2/4p Are Master Regulators of Purine-Starved Cell Transcriptome

Cells use nutrient-dependent intracellular signaling, such as PKA and TOR, to coordinate their metabolism and cell growth with available resources, such as carbon and nitrogen. These signaling cascades converge to several transcription factors, which are then translocated into the nucleus and induce a set of growth or stress response genes. Thus, the specific transcriptional makeup of growing or non-growing cells is set.

The environmental stress response (ESR) is a specific transcriptional program induced by many environmental stresses (heat, oxidation, starvation, etc.). During ESR response in *S. cerevisiae*, upregulated genes for all stresses are genes involved in oxidation proceses and stress signaling. The ESR transcriptional program is mediated via protein kinase A (PKA) and stress responsive Msn2/4p. Genes that are negatively regulated during ESR are connected to the ribosome biogenesis and fermentation. Therefore, when induced, this “transcriptional program” ensures cell survival in multiple environmental stresses [27,40].

As we also see gene expression changes similar to the ESR, we explored whether purine starvation initiates a specific transcriptional pattern. To do that, we analyzed which transcription factors were most probably responsible for the gene transcription pattern induced by purine starvation when compared to exponentially grown cells. When analyzing our expression data with the YEASTRACT tool [41], a set of transcription factors was found most likely to be associated with the upregulated and downregulated genes. Several transcription factors were proposed to be involved in upregulated gene set (*p* < 0.05) Msn2p and Msn4p were transcription factors capable of upregulating 75% and 68%, respectively, of our gene set (see full list with transcription factors in Appendix A). Moreover, our previous research showed that, indeed, Msn2p and Msn4p are involved in purine starvation elicited desiccation and tolerance to thermal shock. When truncating the DNA binding domain of these proteins, cell desiccation tolerance decreases. The decrease in desiccation tolerance is more significant if Rim15p, an upstream regulator of Msn2p and Msn4p, is truncated [42].

Other proposed transcription factors can regulate smaller portion of our gene set, but still are in the agreement of stress resistance phenotype induction. We found stress response related transcription factors Gis1p, Hsf1p, Crz1p, carbon source influenced Mig1p, Hap4p, Cat8p, Adr1p and cell cycle, meiosis dependent Rlm1p, Gat4p, Rme1p, and Yph1p in the list of proposed transcription factors. Effects that would be induced by transcription factors predicted by YEASTRACT correspond to the observed phenotype cells have halted cell cycle, rerouted carbon fluxes, and became resistant to multiple stresses. This allows us to speculate that purine starvation is perceived within the cell, and a coordinated response is launched.

We also compared our data from SD ade− media with publicly available datasets of stationary phase yeasts. Stationary phase cells are an example of where the ESR transcription program is on [30]. We saw similar patterns of gene expression. In both cases, purine-starved *ade8* and stationary phase cells, there was upregulation of heat shock protein genes (*HSP12*, *HSP26*), oxidative markers (cytoplasmic catalase *CTT1*), and hydrophilins essential to overcoming the desiccation–rehydration process (*SIP18* and its paralogue *GRE1*). Therefore, we can now place purine starvation among other stresses capable of inducing ESR.

### 4.4. Intracellular Signalling of Purine Starvation

Pelletier and colleagues used carcinoma cell lines to see how nucleotide depletion affects the cell cycle [43]. The results showed that when purine levels dropped, ribosome assembly was delayed, as nucleotides are needed for RNA. This caused failure of cell cycle checkpoint and p21 accumulation, arresting cells in G1. Overexpression of the *S. cerevisiae* p21 analogue *CIP1* (Ccr4-Not complex inhibitor) also caused arrest in the G1 phase [44]. Yeast cells respond to other environmental stresses, such as hyperosmotic stress, by activating *CIP1* in an Msn2/4p dependent fashion, thus delaying cell cycle [45]. Although *CIP1* was not significantly overexpressed in purine-depleted cells, this does not rule out the involvement of this inhibitor linking purine depletion to G1 arrest.

Hoxhaj and colleagues explored how HeLa cells react to purine synthesis inhibitors [46]. They noticed similar patterns as we did: intracellular concentration of AMP, ADP, and ATP decreased, but cellular adenylate charge stayed constant. They proposed that purines are sensed with the help of the mTOR signaling system, where TSC complex would be responsible for sensing the lack of adenine nucleotides and inhibiting the mTOR pathway further on. Our expression analysis shows downregulation of translation machinery consistent with the involvement of TOR signaling; at the same time, it is necessary to note that *S. cerevisiae* lacks TSC complex [47]. We compared the effect of TOR inhibition by rapamycin with purine starvation on yeast desiccation tolerance (see Figure 5b). We observed that purine starvation induced increased stress desiccation resistance, higher than rapamycin alone. Moreover, purine starvation with simultaneous rapamycin treatment did not increase desiccation tolerance when compared to starvation alone. Therefore, if the TOR system is involved in purine sensing in *S. cerevisiae*, purine depletion signaling is received by TOR somewhere downstream of the canonical rapamycin-sensitive TOR protein Frp1p [48].

## 5. Conclusions

Auxotrophic starvation induces a stress tolerance phenotype in the case of methionine starvation [8] but leads to a stress susceptible phenotype in uracil or leucine starvation [9]. Our results show that purine starvation enriches G1/G0 cells in the culture and “prepares” yeast cells to become multiple-stress tolerant.

We think these distinct phenotypic changes can be explained by one or a combination of several mechanisms acting during purine starvation: a drop in nucleotide content leads to cessation of the cell cycle, accumulation of general stress resistance metabolites trehalose and small heat shock proteins Hsp12p and Hsp26, and firing of Msn2/4-dependent transcriptional program, similar to ESR. The similarity of phenotypes elicited by purine starvation and stationary phase cells by the involvement of similar signaling pathways (Msn2/4p) leads us to conclude that purine starvation indeed elicits ESR phenotype in yeast cells. Moreover, examples from other purine auxotrophic organisms highlight that the ability to survive purine depletion by inducing a stress resistance phenotype might be a universal trait of eukaryotic cells [17].

## Figures and Tables

**Figure 1 jof-08-00029-f001:**
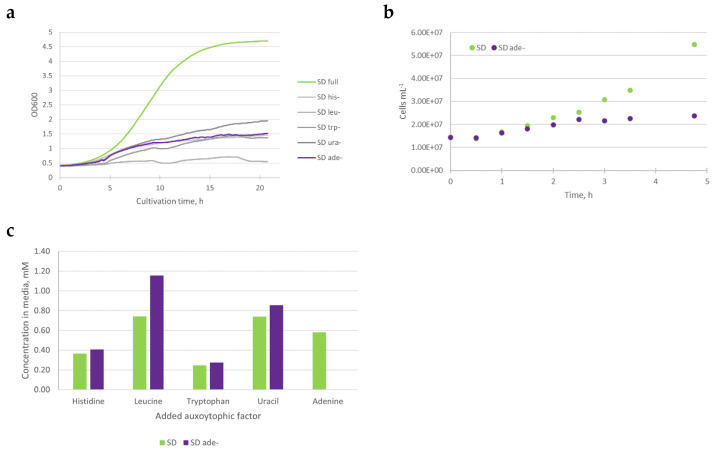
(**a**) Growth of *ade8* (CEN.PK2-1D *MAT alpha his3Δ1; leu2-3_112; ura3-52; trp1-289; MAL2-8c; SUC2 ade8Δ0*) strain in SD media with all auxotrophic factors added in surplus or one auxotrophic factor omitted to starve cells for that particular nutrient. Slight increase in optical density can be observed in most starvation conditions. (**b**) Cell number per mL during first 4 h of growth in SD media or adenine starvation. It can be seen that cells stop increasing in number after 2 h of purine starvation. (**c**) Concentration of auxotrophic factors in growth media after 4 h of *ade8* cultivation. In purine starvation media, all other auxotrophic factors except adenine are still in surplus.

**Figure 2 jof-08-00029-f002:**
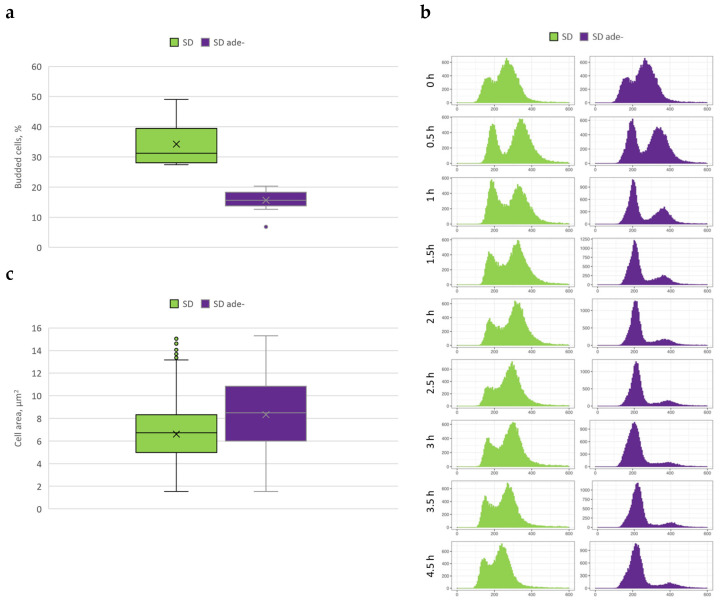
*ade8* cell morphology changes when grown in full or adenine-deficient media. (**a**) Cell budding index. Data from 200 cells analyzed from microscopic images. (**b**) *ade8* strain cell DNA copy dynamics over time in SD and SD ade− media. (**c**) Cell size analysis as determined by area of cross-section in microscopic images. Data from at least 500 cells from each cultivation.

**Figure 3 jof-08-00029-f003:**
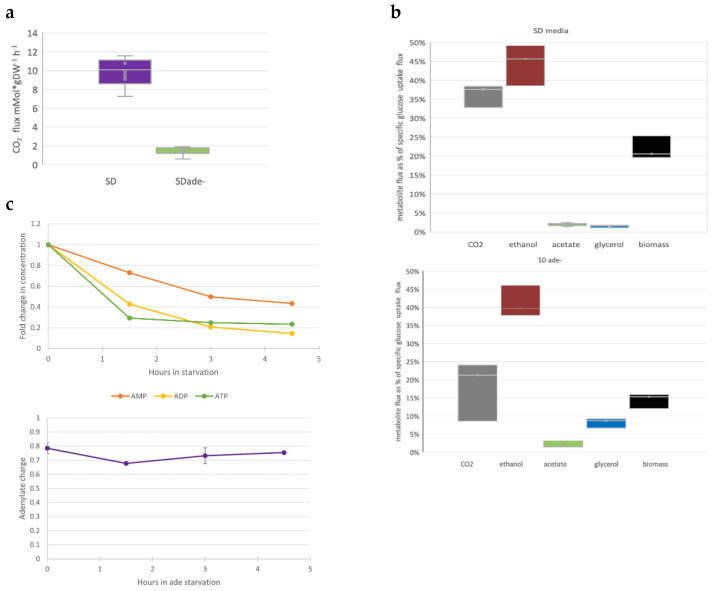
(**a**) Specific CO_2_ production in *ade8* strain cultivation in SD or SD ade− media. (**b**) Carbon flux distribution in *ade8* strain cultivation in full SD media or with adenine omitted. Average CO_2_ flux was measured as produced (mM ∙gDW^−1^∙h^−1^) from SD and SD ade− media. All cultivations were performed in batch mode, in 400 mL Sartorius Qplus fermentation system. Starting volume was 300 mL. CO_2_ was measured by infrared sensor (GasAnalyser, InforsHT). Box plot depicts standard deviations, error bars indicate min and max values from three independent bioreactors. (**c**) Top panel, changes of AXP amount in ade− starved *ade8* cells. Bottom panel, adenylate charge during purine starvation.

**Figure 4 jof-08-00029-f004:**
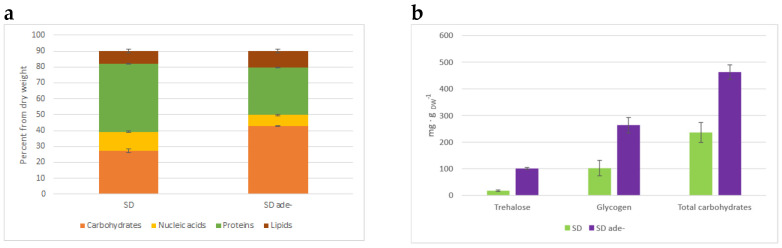
(**a**) Distribution of macromolecules in cell biomass as assessed by FTIR. Data are averages of two biological replicates; error bars show standard deviation among technical replicates. (**b**) Amount of carbohydrates in biomass assessed by anthrone method. Data shown are averages from three biological replicates. Error bars indicate standard deviation of biological replicates.

**Figure 5 jof-08-00029-f005:**
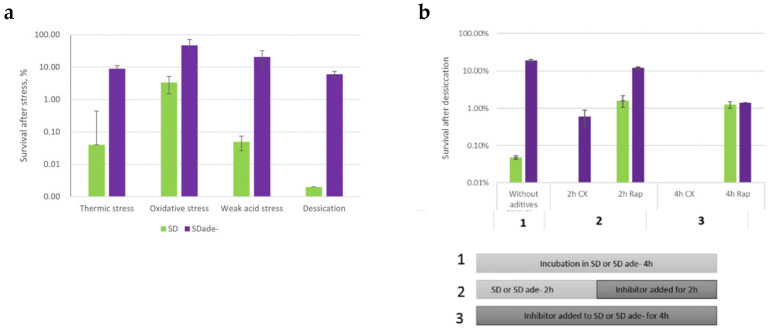
*ade8* strain stress resistance after cultivation in SD or SD ade− medium. (**a**) Cell survival after exposure to sublethal stress conditions. *ade8* cells were cultivated in SD and SD ade− for 4 h, then exposed to heat shock (53 °C, 10 min), oxidative shock (10 mM H_2_O_2_, 50 min), acetic acid stress (plated on YPD with 0.1 M acetic acid, pH 4.5), or desiccation (30 °C for 6 h) after stress treatment; cfu∙mL^−1^ OD^−1^ was assessed by plating on YPD plates. (**b**) Cell survival after desiccation and addition of cycloheximide (35 µg∙mL^−1^) or rapamycin (250 µg∙L^−1^) during starvation (top panel); setup of experiment (bottom panel). Cells were treated with cycloheximide or rapamycin at various time points during adenine starvation and then desiccated. CFU plating was done as in other stress assays.

**Figure 6 jof-08-00029-f006:**
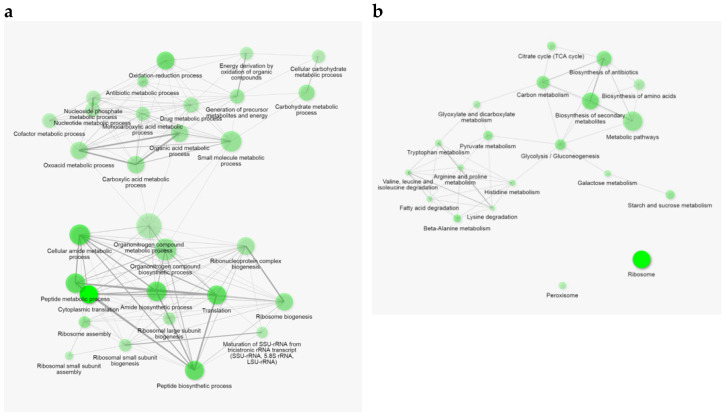
Gene pathway enrichment analysis, in which transcripts significantly up- or downregulated (−2 > log_2_FC > 2) when compared to *ade8* cells cultivated in synthetic complete media were selected. For those genes, enrichment analysis was done and plotted using ShinyGO v0.61 [29]. Plot also shows relationships between enriched pathways. Two pathways (nodes) are connected if they share 20% (default) or more genes. Darker nodes are more significantly enriched gene sets, bigger nodes represent larger gene sets, and thicker edges represent more overlapped genes. (**a**) Enrichment in GO metabolic process terms; (**b**) enrichment in KEGG pathway terms.

**Figure 7 jof-08-00029-f007:**
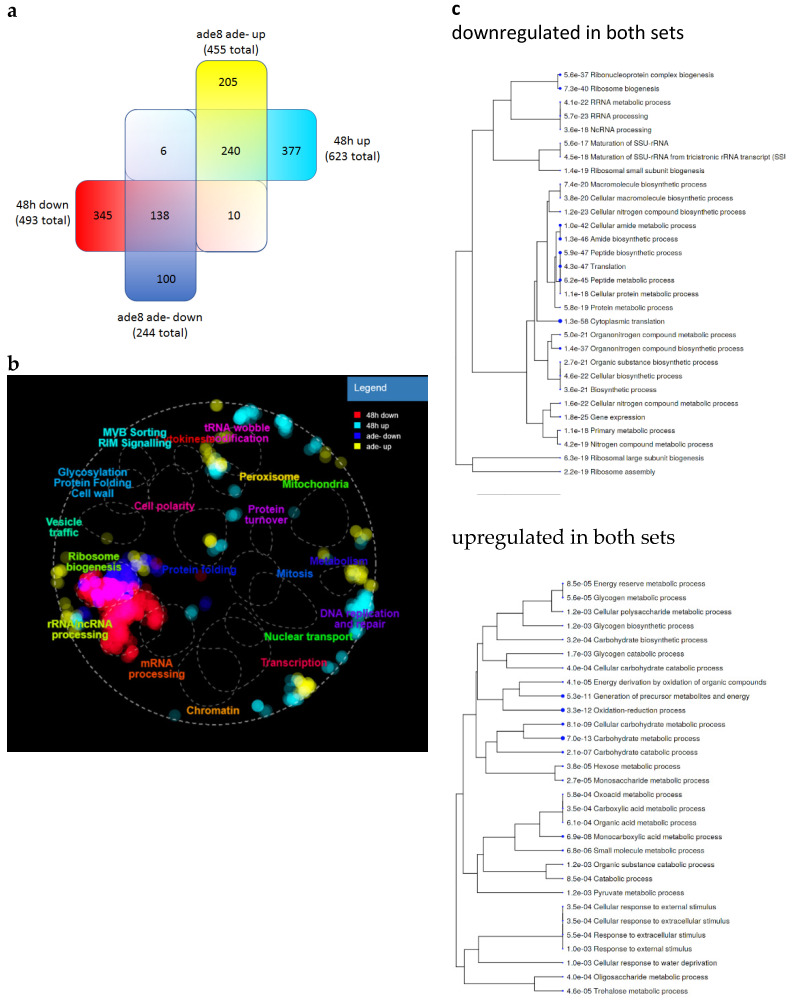
Comparison of gene expression of *ade8* strain in SD ade− media after 48 h and JPY10I strain (*MATa/*α *ura3*Δ*0*/*ura3*Δ*0 leu2*Δ*0*/*leu2*Δ*0 lys2*Δ*0*/*lys2*Δ*0 ADE2*/*ade2*Δ*::hisG HIS3*/*his3*Δ*200*) after 48 h growth in full media (Geodata set GSE111056). Gene sets were obtained by comparing gene expression during starvation and stationary phase with expression of respective strain during exponential phase. Significantly (−2 > log_2_FC > 2) up- or downregulated genes were further analyzed. (**a**) Number of common genes that changed their expression when comparing purine-starved and stationary phase cells. (**b**) Same genes plotted with TheCellMap.org. (**c**) GO term enrichment analysis of upregulated genes in purine-starved and stationary phase cells.

## Data Availability

The expression dataset was submitted to the European Nucleotide Archive (ENA) database, accession no. PRJEB40525.

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
