# Peer review of "Purine Auxotrophic Starvation Evokes Phenotype Similar to Stationary Phase Cells in Budding Yeast"

_jof, 2021, doi:10.3390/jof8010029_

Round 1

Reviewer 1 Report

In Purine Auxotrophic Starvation Evokes Phenotype Similar to Stationary Phase Cells in the Budding Yeast, Kokina and co-authors show that starving budding yeast cells of purines, in this case, adenine, is bad for them. Using a variety of analyses, the researchers show that cells that can not make their own adenine, via the uncommon ade8 allele, possess traits of cells that are in stationary phase, i.e., cells that are starved. Unfortunately, I can not understand the purpose of this study, since the fact that budding yeast cells will eventually consume all the nutrients they are provided is fundamental common sense that is covered in every lab manual on basic budding yeast husbandry. It is exactly why budding yeast cultures are grown in media that contains vast excesses of nutrients they may not even need, and why any serious budding yeast researcher only uses cells from cultures in logarithmic growth in their experiments. Additionally, the manuscript is extremely poorly written; it was difficult to read. Thus, I can not recommend publication in the Journal of Fungi.

Author Response

Answer to the reviewer

Thank You for the comment.

Indeed, when cultivated in the laboratory settings, yeasts are provided with balanced, nutrient rich broth and their uninterrupted growth is observed. However, in the “wild” yeasts seldom encounter “balanced” and nutrient rich growth media. Typically, one essential nutrient (carbon/ nitrogen/ phosphorous/ sulfur, etc.) gets depleted, yeast cell proliferation ceases and specific starvation for that nutrient sets in. S. cerevisiae is well adopted to “natural” starvations. Since these starvations coincide with phenological events and annual cycle (sugar rich fruit development, onset of winter etc.), “natural” starvations induce strong environmental stress resistance phenotype.

Although many auxotrophies are introduced within the yeast strains for gene engineering, there are many “naturally occurring” auxotrophies found within the “wild organisms”. For example, many yeasts are biotin, inositol auxotrophs (reviewed by Perli et al., 2020, 10.1002/yea.3461). Up till now other authors have dissected physiology of uracil, leucine, methionine auxotrophies and their specific starvations effects on yeast phenotype (Petti et al, 2011, 10.1073/pnas.1101494108,  Boer et al., 2008, 10.1073/pnas.0802601105, Santos, et al., 2021, 10.1007/s11357-020-00265-2, Green et al., 2020, 10.1371/journal.pbio.3000757)

Interestingly, single cell eukaryotic intracellular parasites (Leishmania spp., Toxoplasma spp., Plasmodium spp.) are purine auxotrophs (de Koning, et al., 2005, 10.1016/j.femsre.2005.03.004).

In one of our previous studies (Kokina et al. 2014 10.1111/1567-1364.12154), we demonstrated, that purine auxotrophic yeast (W303 ade2 strain) in rich (yeast extract – bacto peptone based media) rapidly depletes purine and subsequent purine starvation starts. While starving for purines, W303 ade2 strain cells became larger, desiccation tolerant and population budding index decreases.

In the current JoF-1507672 manuscript we dissected purine starvation effects on cell metabolism (cell cycle, central carbon metabolism, macromolecular content, stress resistance, transcriptomics) in more detail. Our results demonstrate that budding yeast can withstand deficiency of purine and it successfully reacts to it by turning on general stress resistance - to preserve the integrity and functionality of the cell. This contrasts with other auxotrophic starvations: uracil or leucine depletion induce futile glucose consumption, cells can not complete cell cycle, their chronological life span decreases (Boer et al., 2008, 10.1073/pnas.0802601105). Moreover, Leishmania (purine auxotroph) when starved for purine also induce stress resilience phenotype (Martin, et al., 2014, 10.1371/journal.ppat.1003938).

To summarize, we think, that our results are interesting and merits publication in JoF because of three reasons:

  • purine depletion induces strong stress resistance phenotype in S. cerevisiae.
  • many phenotypic and molecular traits of purine starved yeast resemble stationary phase cells
  • our results complement observations in other purine auxotrophic organisms and demonstrate, that similar conditions can be replicated in well-established model organism budding yeast.

Thank You for the note on language quality. We performed language correction offered by JoF and we hope, that quality of English has improved in the revised manuscript.

We hope that we have addressed your concerns about our manuscript

Reviewer 2 Report

Dear editor, 

Dear authors, 

The manuscript presented by Agnese Kokina and her collaborators emphasises important efforts to study the effects of purine starvation in the budding yeast. They concluded that the purine auxotrophic starvation induces phenotype distinctive from exponential cells and at the same time induce accumulation of metabolites capable to increase stress tolerance. 
In order to raise the value of the paper, I highly recommend that the discussion about the transcriptional factor Msn2/4p, the master regulators of purine starved cells transcriptome, be extended to other important transcriptional factors involved in stress response, such as Yap1p, or Kcs1 and Vip1p. 
General observations:
Even the manuscript is well presented, the methods and results are well described, it is necessary a serious revised of English language. 
For the entire manuscript, the authors should use the same notation for figures, Figure x or Fig X. 
In all the manuscript ade8 should be written using italic font. 
The formulation ox-red processes can be replaced with redox process.
Some minor points to be resolved:
Line 26. Saccharomyces cerevisiae should be written using the Italic font. 
Line 264. Please explain the phrase, in terms of concentrations: Additionally, CO2 produced in purine starved cells is equal to ethanol produced only. 
Line 459. Remove double folate. 
Line 596. Please insert the number of the reference. 
Figure 5. The authors should mention the concentrations of cycloheximide and rapamycin.

Author Response

Answer to the reviewer 2

Thank you for reading our manuscript and helpful suggestions on improving it.

 According to your suggestion, we have extended the discussion part 4.3. to delve more on putative transcription factors involved in gene expression changes observed during purine starvation. [see lines 586-615.]

When analyzing transcriptome data Yap1 did come up as one of the transcription factors that regulates 80% of the genes that are upregulated (log2(FC)>2) in our data set, but the p value when compared to the whole S.cerevisiae genome was extremely high 0.95 so we decided not to include it in our manuscript. As for the Inositol phosphate kinases that you are proposing - that is certainly an interesting idea, but unfortunately, we are not able to assess the involvement of secondary messengers with the tools we are using. GO term analysis of transcriptome that is upregulated and downregulated during purine starvation (-2>log2(FC)>2) does not highlight involvement of inositol kinases and also when performing gene search, we came up with only one gene involved in inositol phosphate metabolism - namely INM1 that was downregulated log2FC=-3 in our data set. At the same time, we are aware that transcriptomics is not very suitable to elucidate involvement of secondary messengers, so we decided to leave it out of the manuscript but it does make an interesting field of future studies.

For general observations:

We have employed the language editing services offered by JoF and we hope that language quality is now improved.

For remark on line 264 

The idea behind this sentence was, that when we compared amount of the CO2 produced in purine starved and normally growing yeast, we saw, that when starved for purine CO2 released is equimolar to ethanol produced.

To clarify the sentence, we rewrote it like that:

Line 272-273: Additionally, amount of CO2 released from the purine-starved cells was equimolar to the ethanol produced.

We have added concentrations of cycloheximide and rapamycin in the legend of the Figure 5

We have also implemented all your suggested changes according to formatting – same style for Figure X, italics, reference in line596 (now line 631), correct spelling of redox processes, removed double folate.

We hope that changes are according to your suggestions and we have improved the quality of the manuscript.

Reviewer 3 Report

Overall interesting study but needs some improvement on the writing.

  1. line 44-45 - some problems with reference
  2. Please do not use * as multiplication sign
  3. Line 84 - please use full name of SD medium
  4. Line 91 please remove +
  5. Line 99 - it should be D2O
  6. Line 578 - please use names with some references
  7. Please use italics for microorganisms names in references section

Author Response

Thank you for reading our manuscript and helpful suggestions on improving it.

We have employed the language editing services offered by JoF and we hope that language quality is now improved.

 We have implemented the changes that you suggested:

Changed the structure of sentence to be clearer on reference [lines 46-48]

Corrected the multiplication sign throughout the manuscript

Included full name of SD media

Corrected the temperature and D2O notations

We have italicised all the names of microorganisms in references.

On the remark of reference for line 578, we assume that you were referring to the line 587 and we have now changed the structure of the sentence to include names of the authors. [see lines 640 and 650]

We hope that changes are according to your suggestions, and we have improved the quality of the manuscript.

Round 2

Reviewer 1 Report

I thank the authors for their response and regret my initial harshness in the first review. I am swayed by the authors' explanation in the response, and I think it would improve the manuscript to include this or a variation of this rationale in the introduction or the discussion. 

Reviewer 2 Report

Dear Editor, 

The authors responded to my proposals in a satisfactory manner. I appreciate their work and I recommend publishing the manuscript in its present form.